# Microrollers flow uphill as granular media

Samuel R. Wilson-Whitford[1], Jinghui Gao [1], Maria Chiara Roffin[1,2], William E. Buckley[1] & James F. Gilchrist [1] ✉

Pour sand into a container and only the grains near the top surface move. The collective motion associated with the translational and rotational energy of the grains in a thin flowing layer is quickly dissipated as friction through multibody interactions. Alternatively, consider what will happen to a bed of particles if one applies a torque to each individual particle. In this paper, we demonstrate an experimental system where torque is applied at the constituent level through a rotating magnetic field in a dense bed of microrollers. The net result is the grains roll uphill, forming a heap with a negative angle of repose. Two different regimes have been identified related to the degree of mobility or fluidisation of the particles in the bulk. Velocimetry of the near surface flowing layer reveals the collective motion of these responsive particles scales in a similar way to flowing bulk granular flows. A simple granular model that includes cohesion accurately predicts the apparent negative coefficient of friction. In contrast to the response of active or responsive particles that mimic thermodynamic principles, this system results in macroscopic collective behavior that has the kinematics of a purely dissipative granular system.

When passive granular matter is poured onto a substrate, it forms a heap of material consisting of a near surface flow of grains and an underlying pile of nearly static particles[1–3] (Fig. 1a). Within this flowing layer, grain motion is correlated as it transfers potential energy into translation and rotation and eventual frictional dissipation through multibody collisions[4]. This flowing layer is generally characterized by its angle of repose, $\theta$, which is related to the friction interactions of the particles. This trivial dinner table experiment is analogous to a wide range of natural phenomena such as avalanches and dune formation and is ubiquitous with industrial processes for powder handling that follow scaling laws related to their rheology[5].

Rather than letting particles passively fall, heap, and flow down an incline driven by gravity, we explore a system where energy is input at the constituent-level through magnetic activation of torque on each particle. This is coupled with magnetically-tunable attractions that alter their interparticle interactions. When activated, a dense bed of these microrollers spontaneously generates a steady heap against a static wall. This is a result of an uphill flowing layer characterized by a negative angle of repose. Grains are recirculated through the underlying bed (Fig. 1b). This negative dynamic angle of repose is not to be confused with negative static angles of repose measured in cohesive or

interlocking granular packings[6]. For stronger magnetic interactions, rather than imparting stronger cohesion[7], the entire bed is fluidised by overcoming the weight of the bed and breaking the static force chains associated with a granular heap[8]. The experimental realisation of this system, where granular flow is driven by torque imparted at the particle level and friction, results in the emergence of collective dynamics that scale as gravity driven granular flows.

## Results and discussion

The microrollers used in this study are non-colloidal Janus particles synthesized from polydisperse polymethyl methacrylate microbeads of radius $19\,\mu m < a < 26\,\mu m$ (Supplementary Fig. 1) and half coated with a 100 nm layer of iron using physical vapour deposition (PVD)[9] (Supplementary Fig. 2). The microrollers are dispersed in ethanol where the iron cap undoubtedly becomes iron oxide and has a weak, single vector, off-centre permanent dipole with poles located at opposite points along the edge of the cap. The phenomenon described herein has also been observed in air, with no substantial change in the observed behaviour, but advantageously the use of a viscous fluid instead of air is helpful in avoiding electrostatic build-up on the particles or the container surface. The emergence of self-

[1]Department of Chemical and Biomolecular Engineering, Lehigh University, Bethlehem, PA 18015, USA. [2]Department of Physics, School of Science and Technology, Nottingham Trent Univeristy, Nottingham NG11 8NS, UK. ✉e-mail: gilchrist@lehigh.edu

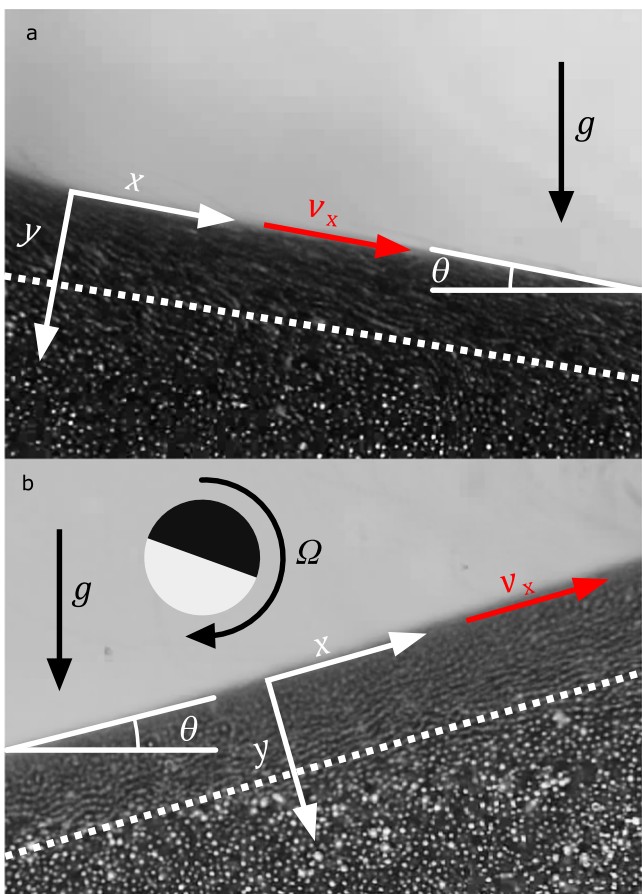

**Fig. 1 | Gravity-driven and magnetically-driven flowing layer of ferromagnetic Janus particles.** Intensity average images of (**a**) a gravity driven flow in a granular heap of unactuated Janus particles and, in contrast, (**b**) an uphill flow of the Janus microrollers driven by magnetic actuation, including an illustration of the direction of particle rotation. Movies of uphill granular flow are available (see Supplementary Information). The relative magnetic field strength is $(\beta/\beta_O)^2 = 3.5$ and the granular bed depth is $\Delta/2a = 26.0$. The dotted white line is an approximate representation of the flowing layer.

induced by the applied field, where $F \propto \beta^2$, as seen in the literature.[12] Therefore, magnetic field is scaled as $(\beta/\beta_O)^2$, where $(\beta/\beta_O)^2 = 1$ generates no substantial particle motion. Note that dipole interactions are not referring to those typically seen on the colloidal scale, but instead refer to the macroscopic dipole of the magnetic material forming the cap of each particle. The impact of increasing the effective interparticle particle friction by increasing magnetic strength, $(\beta/\beta_O)^2 > 1$, is detailed below. The frequency of magnet rotation, $\Omega$, is chosen within the wide range of rotation rates for which the observed phenomena do not deviate greatly from that reported herein. Also important, the rolling velocity of a single microroller in dilute conditions is independent of the field strength (Supplementary fig. 6)[13]. Movies of uphill heaping are available (see Supplementary Movie V1–V4). Samples of different relaxed state bed depths were used and denoted as a diameter normalised depth, $\Delta/2a$.

From a cursory view of this system, the kinematics of the granular motion of these particle rollers is almost identical to that of a typical gravity driven steady avalanche in a granular system, yet reversed to flow uphill. The particle bed demonstrates a steady-state flowing layer with decreasing particle velocity perpendicular to the surface. In some respect, this system mimics the continuous flow displayed in rotating drums[14–16], in a regime most similar to rolling/cascading where there is no flow intermittency, where particles are continuously leaving the flowing layer at one end and are slowly transported back to the opposite end. In a rotating drum, this occurs by the solid body motion of particles with the drum below the flowing layer, which are stationary in this experiment. Alternatively, comparisons could be made to fluid-sheared granular transport, such as that presented by Houssais, showing behaviours present in both the bed load and creeping regimes[17]. In the presented system, recirculation of particles occurs through a backflow of material in the fluidised region below the uphill flowing layer. Additionally, average velocities in the bulk of the heap are an order of magnitude smaller than that in the flowing layer. Particles are not rotating in perfect synchronicity with the magnetic field due to the stress imposed by the weight of the granular bed and the friction resulting from multiparticle interactions. In the flowing layer particles experience less stress from the weight of particles above and their rotation rates are generally more synchronized with the magnetic rotations, with fluctuations resulting from collisional interactions changing the motion of these particles. Successive experiments were performed on relatively shallow to increasingly deep granular beds, measured as $\Delta/2a$ at rest in a uniform bed prior to particle motion.

An estimation of the friction of granular systems is traditionally determined by the Mohr-Coulomb yield criterion by measuring the static or dynamic angle of repose, where the yield criteria is given by

$$\mu = \tau/\sigma = \tan(\theta) + c/\sigma, \tag{1}$$

where $\mu$ is coefficient of friction, $\tau$ is the shear stress, $\sigma$ is the normal stress, and $c$ is particle cohesion. Without any applied magnetic field, these particles flow freely without any apparent cohesion in a wide drum with an angle of repose of $\tan(\theta) \approx 0.45$ which is a direct measure of particle friction. The dynamic angles of repose of microrollers, plotted as $-\tan(\theta)$ vs. $(\beta/\beta_O)^2$, display two distinct regimes (Fig. 2). For $(\beta/\beta_O)^2$ just above 1, these samples have vanishingly thin flowing layers and take a very long time to reach steady state (Supplementary Movie V1). The particles beneath the flowing layer are quasi-static, minimizing the degree of sub-flowing layer recirculation. The flow eventually stagnates, where the field strength is insufficient to overcome the potential energy to flow uphill, resulting in large negative values of $\tan(\theta)$. Experiments with increasing values of $(\beta/\beta_O)^2$ causes the magnitude of $\tan(\theta)$ to reach a minimum negative value $(\beta/\beta_O)^2 \approx 9$. This is a result of the increased fraction of mobile/fluidised particles as the fluidised fraction $\chi \to 1$ (Supplementary Fig. 7). Upon this transition, the flowing layer is thicker and recirculation/backflow

organized structures that mirror thermodynamic properties[10] and instabilities[11] have been characterized in dilute, quasi-2D microroller systems that differ significantly from this dense 3D flow. Without imparted magnetic torque, a dense bed of these particles flows under the influence of gravity similar to unfunctionalized microbeads. A set of rotating permanent magnets is mounted on a rotating wheel horizontal in the plane and perpendicular to the sidewall of the cuvette and subsequently positioned below the cuvette. The magnetic field imparts both the particle-level torque and the interparticle cohesion influencing the degree of friction. The magnets are much wider than the 1 cm² square-bottomed cuvette holding the particles and spaced out such that the particles experience a relatively uniform field across the container, resulting in a near sinusoidal modulated magnetic field as each magnet rotates past the bottom of the cuvette. The amplitude of the magnetic field is a function of the distance of the particle bed above the rotating magnets, $h$, where the field strength varies proportionately as $\beta \propto 1/h^2$, (See Supplementary Figs. 4 and 5). A minimum field strength exists, $\beta_O$, where the weaker magnetic field generates no discernible particle motion and the particle bed is essentially static, in this case when $h = 60$ mm. Below $\beta_O$, the magnetic torque felt by individual particles is insufficient to overcome static friction and the force network within the particle bed. Interparticle interactions are influenced by magnetic dipoles

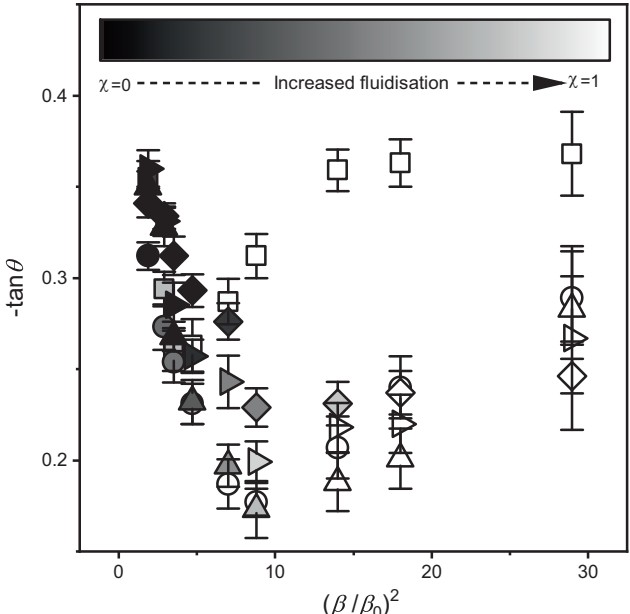

**Fig. 2 | Negative angle of repose in uphill heaping.** Dynamic Angle of Repose quantified as $-\tan(\theta)$ as a function of relative magnetic field strength, $(\beta/\beta_0)^2$, for samples $\Delta/2a$, 9.5 (square) 18.5 (circle) 26.0 (upward triangle) 31.0 (sideward triangle) 39.0 (diamond). Logarithmic colourbar indicates the fraction of particles with non-zero velocity, $\chi$. Error bars are the standard deviation over multiple trials. Error bars represent the standard deviation of the angle of repose in each case.

under the bed balances the uphill convection to reach a dynamic steady state of uphill and recirculation flux (Supplementary Movie V2). By increasing $(\beta/\beta_0)^2$ beyond the point where $\chi = 1$, the magnitude of the uphill dynamic angle of repose increases. This transition coincides closely with the point at which the degree of fluidisation in the bed is 1 (Supplementary Movie V3), at $(\beta/\beta_0)^2 \sim 9$, as indicated by the greyscale colourmap of the data points in Fig. 2. In this regime, there is no network of force chains and the angle of repose does not simply represent friction or cohesion, but the pressure gradients generated within the recirculating bed, distinctively different from the dynamic angle of repose that is generated by flowing passive granular media. The trend is the same for all bed depths with the exception of $\Delta/2a = 9.5$, where the fluidised layer formed has a thickness equal to that of the depth of the bed at $(\beta/\beta_0)^2 \sim 2.9$–3.5 where the stress associated with the particle-wall interactions at the bottom of the bed likely play a more significant role. Movies of the fluidisation fraction and flow motion for $(\beta/\beta_0)^2 \ll 9$ (Supplementary Movie V1), $(\beta/\beta_0)^2 < 9$ (Supplementary Movie V2) and $(\beta/\beta_0)^2 > 9$ (Supplementary Movie V3) are shown in the Supporting Information.

To characterise the kinematics of the granular flowing layer flow that arises from the collective motion of these microrollers, particle image velocimetry (PIV) of steady-state heaping was performed at varying field strengths, $(\beta/\beta_0)^2$ and bed depths. The velocity measured at the midpoint of the free surface is scaled by single particle rotation rate, $v_x/2a\Omega$, and plotted against the normalised depth into the sample, $y/2a$, with the uphill direction defined as positive values of $v_x$ (Fig. 3a–e). For all sample depths we observe high shear rate near-surface velocity profiles similar to that measured in gravity driven flows[2]. The decrease from the maximum velocity near the surface is linear with a constant local shear rate that decays as the depth increases. The flow recirculation, observed in the supporting movies is also seen in the velocity profiles in Fig. 3, represented by backflow with a negative velocities, $v_x < 0$, relative to the heaping direction and is more significant for shallow bed depths where the entire bed is fluidised.

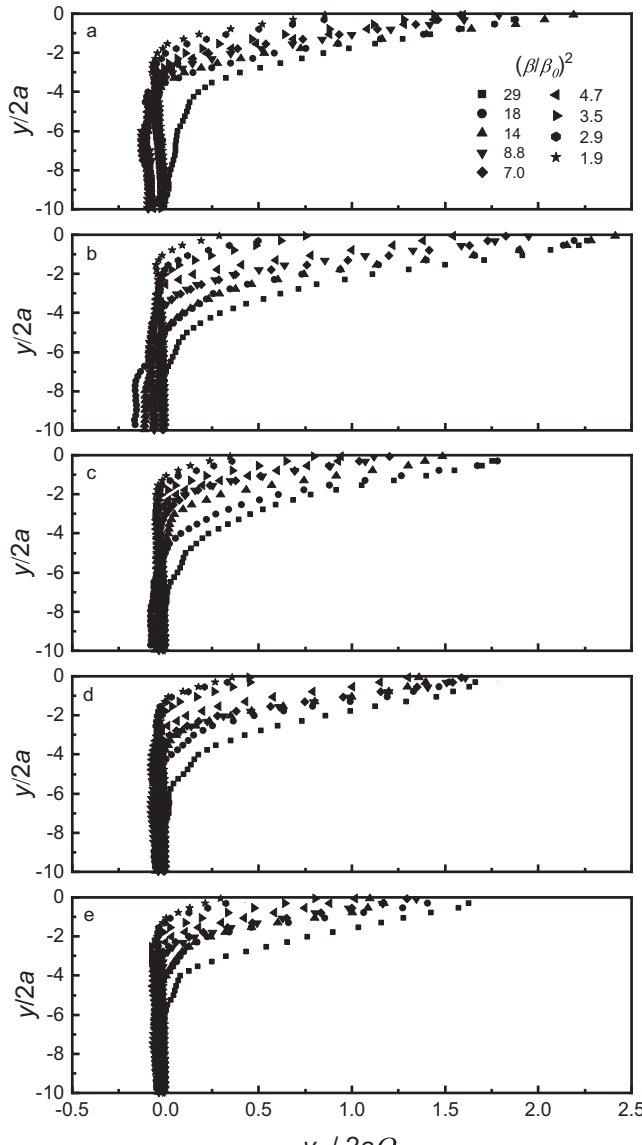

**Fig. 3 | Velocity profiles of uphill heaping perpendicular to the flowing layer.** Velocities (uphill, $v_x > 0$) of particles in magnetically agitated dynamic heaps as a function of the diameter normalized depth into the sample ($y/2a$), perpendicular to the free surface for samples with a resting, pre-fluidised normalized depth of $\Delta/2a = 9.5$ (**a**) 18.5 (**b**) 26.0 (**c**) 31.0 (**d**) 39.0 (**e**). All samples are measured over a range of magnetic strengths, $(\beta/\beta_0)^2$.

For gravity driven flows, it is common to scale the velocity by the driving force, $(gd)^{0.5}$ [2,5] The driving force in this study, localised torque introduced at the particle scale, and for microroller "critters" has been demonstrated as a linear combination of rotation rate producing a constant local shear, similar to that observed in Fig. 3. In this case, the dimensionless shear rate,

$$\dot{\gamma} = \frac{1}{\Omega}\frac{dv_x}{dy} \tag{2}$$

captures this relatively simple phenomenon even though the individual particle motion is far from lock-step rotation with the rotating magnets. PIV data from Fig. 3 at all sample depths, $\Delta/2a$, and all field strengths, excluding the strongest magnetic field, $(\beta/\beta_0)^2 = 29$, were collapsed with $\dot{\gamma}$ (Fig. 4). Note that the data from the strongest field is omitted from the collapse. At this field strength, increased chaining in

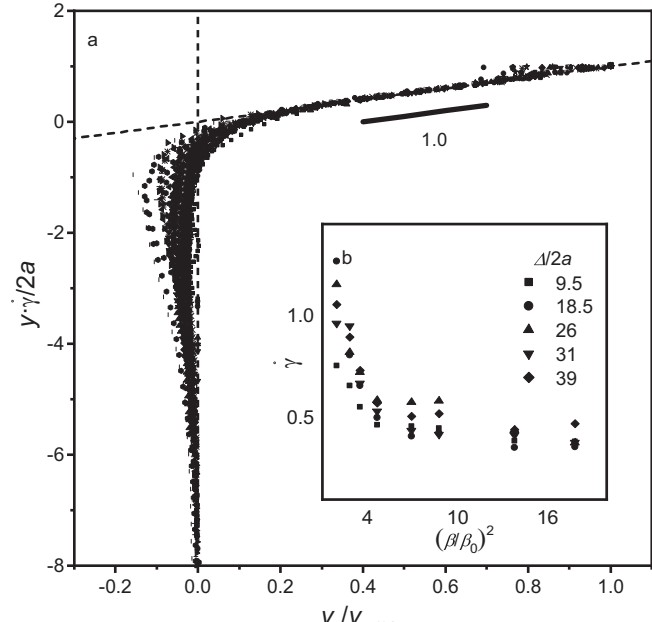

**Fig. 4 | Velocity profiles collapsed by dimensionless shear rate. a** Collapsed curves from all bed depths $\Delta/2a$ and all magnetic field strengths $(\beta/\beta_0)^2$. **b** Inset shows relationship between dimensionless shear rate, $\dot{\gamma}$, and field strength. The vertical dotted line marks the uphill vs. recirculation point where the velocity reverses its direction. The dotted line that fits the data for larger velocities in the linear regime intersects with $v_x/v_{x,\,max} = 0$ to highlight how the data is shifted for this analysis.

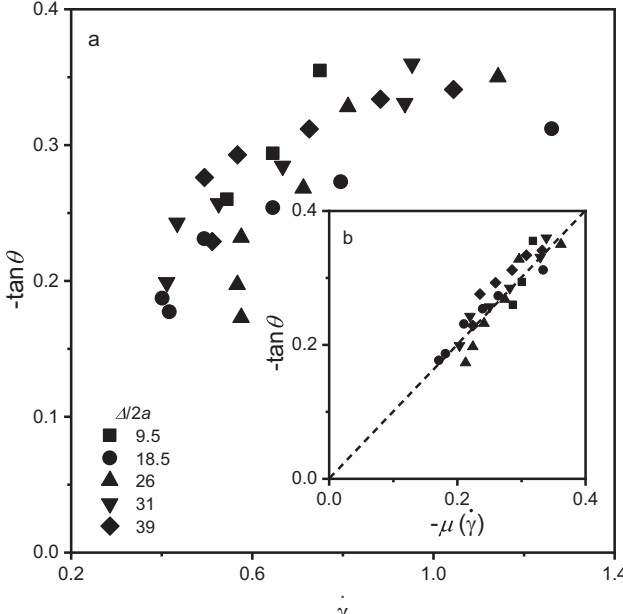

**Fig. 5 | Negative angle of repose as a function of the shear rate and the coefficient of friction. a** The angle of repose, $-\tan(\theta)$, plotted as a function of shear rate, $\dot{\gamma}$. **b** Inset shows the angle of repose as a function of and the coefficient of friction, $\mu(\dot{\gamma})$. The angle of repose and shear rate in experiments where $\chi < 1$ follows a general upward trend that is accurately predicted by calculating a coefficient of friction that accounts for field strength dependent cohesion. The dashed line is $\mu(\dot{\gamma}) = \tan(\theta)$, which tracks very close to the best fit line with slope of 1 and y-intercept of $-0.0087$.

the particle bed leads to a non-flat interface of chained particles moving uphill, suggesting an upper limit to the steady state behaviour. In contrast, at the low field limit, the collapse follows until the flowing layer is just a couple of particles thick. The collapsed data, shifted vertically such that the linear regime intersects the extrapolated such that $v_x/v_{x,\,max} = 0$ at $y\dot{\gamma}/2a = 0$, shows consistent linear behaviour across all sample variations, with a slope of $1.02 \pm 0.02$ within the flowing layer. This slope directly equates the individual particle rotation and rolling to the resulting shear rate. The particle velocity decreases in a linear fashion into the bed away from the free surface until the magnitude of the velocity is similar to that of the reverse flow that exists under the flowing layer. The consistency with which the different curves collapse shows that, just like a gravity-driven granular flow, this uphill flow can scale with its driving force, in this case the rotation rate. As opposed to gravity-driven granular flows where dissipation at the particle scale has a large heterogeneity and collapse of such data is more approximate, here particle rotation is generated locally, and it is unexpected that the result scales as a granular medium.

From Fig. 4b we see that dimensionless shear rate is a function of magnetic field strength. It has a sharp decline as $(\beta/\beta_0)^2$ increases for low field strength and plateaus to $\dot{\gamma} \approx 0.5$. This result is at first counterintuitive. A weaker field strength, where interparticle magnetic interactions are weaker, results in a higher shear rate suggesting more efficient transport or imparting a greater degree of stress[18]. For small $(\beta/\beta_0)^2$, the flowing layer thickness is very small, thus transporting very little material uphill. This analysis is for the top-most layer of flow rather than the transition regime deeper within the granular bed where normal stress increases, eventually resulting in stagnation deep in the granular bed. These particles at the free surface experience essentially no granular pressure from above and can move in a more synchronized motion with the magnet. Combining the results of Fig. 2 and Fig. 4 for systems that are not fully fluidised, $\chi < 1$, enables analysis of the impact of cohesion imparted by the magnetic field. Prior work[19] has

developed a constitutive equation that relates the coefficient of friction to the shear rate,

$$\mu(I) = \mu_{max} + (\mu_{max} - \mu_{min})/\left(\frac{I_0}{I} + 1\right), \tag{3}$$

where $\mu_{max}$ and $\mu_{min}$ are the maximum and minimum angles observed, $I$ is the inertia number $I = \dot{\gamma}2a/\sqrt{\sigma_{yy}/\rho}$ scaling the shear rate with particle size, surface pressure and density. $I_0$ is a fitting parameter for the observed plateau in the coefficient of friction. In the topmost regime of the flowing layer where changes in $\sigma_{yy}$ are small and $\rho$ is roughly constant, $\mu(I)$ rheology simplifies to a Bagnold profile with a linear dependance on $\dot{\gamma}$. This model alone does not capture the relationship between the angle of repose and shear rate, shown in Fig. 5a. With addition of cohesion that scales with the magnetic field strength, the coefficient of friction is

$$\mu(\dot{\gamma}) = \mu_{max} + (\mu_{max} - \mu_{min})\dot{\gamma} + \alpha(\beta/\beta_0)^2, \tag{4}$$

where $\mu_{max}$ and $\mu_{max}$ are the maximum and minimum angles observed for each depth. The lone fitting parameter in this system, $\alpha$, is used to scale the magnitude of cohesion due to dipole-dipole attractions, proportional to $(\beta/\beta_0)^2$. Figure 5b demonstrates a quantitative prediction of the measured angle of repose to the predicted coefficient of friction for $\alpha = 7.6 \times 10^{-3}$. Cohesion is necessary to overcome lubrication force imparted between freely rotating particles. Similar to active systems that can demonstrate negative viscosity[20], this system has a negative coefficient of friction. Energy is injected at the microscale, resulting in local shear that is dissipated in the resulting kinematics that follow granular behaviour.

Alternatively, with a higher field strength, the flowing layer is deeper and there is more local dissipation through frictional contacts. The work associated with the flow of these rolling particles is larger

because of the weight of the granular media through the flowing layer. The plateau of $\dot{\gamma} \approx 0.5$ may relate to the fact that these Janus particle microrollers, that have magnetic dipoles only, will have a finite amount of surface area of particle-particle interactions where interparticle interactions are influenced by the magnetic dipoles. The growth of chains changes the dynamics of the system significantly and scaling by particle diameter may not be appropriate. In the dense granular bed, the fast exchange of particle contacts is highly complex and a deeper statistical analysis of particle interactions may be necessary to characterize how energy is transferred from the particle-level throughout the bed.

In summary, the emergent behaviour observed from imparting torque on individual particles is the granular behaviour as is expected in gravity driven flows. A rotating set of magnets imparts localised torque that results in particle rotation. Importantly, this suggests that the responding macroscale stress and related kinematics of a broader class of responsive systems on the granular scale could be described using the insights already obtained for gravity-driven flows. This localised control of particle scale torque allows for a robust set of experimental controls of granular behaviour that could only be considered in simulations where fictitious physical forces could be imparted on the system. The negative coefficient of friction that includes cohesion accurately predicts the negative angle of repose. This result also shines light on the broader consideration of local vs. nonlocal stress in granular media[21,22]. The observed flowing layer arises from imparting internal stress conditions at the particle-level rather than large-scale stresses from body and surface forces on the bulk and boundaries of the granular bed. This suggests an ergodic set of reversed cause-effect relationships could exist between driven and dissipative systems[21].

# Methods

## Materials
38−53 μm (measured; d[3,2] 43.6 ± 5.9 μm) polymethyl methacrylate microparticles were purchased from syringia lab supplies. 190 Proof ethanol was purchased from Fisher.

## Equipment
4 ×1000 G neodymium magnets (60 ×10 × 3 mm) were purchased from MIKEDE. Imaging was performed with a USB Jiusion WIFI Digital Microscope F210. Programmable Lego-EV3 and motor. A PTFE stoppered Cuvette (21FLG10−Macro Fluor 21 Optical glass 3.5 mL) was purchased from FireFlySci. Kindle E-reader purchased from Amazon. Table-mounted supports and brackets were purchased from ThorLabs. Magnetic field strength was measured with a Senjie SJ200 Teslameter, accuracy 1%, precision 0.1 mT.

## Janus particle synthesis
Sub-monolayer films of 38−53 μm (measured; d[3,2] 43.6 ± 5.9 μm) diameter PMMA particles were prepared on a roll-to-roll scale by automated Langmuir-Blodgett[23]. The dried films were coated with 100 nm of Fe by physical vapour deposition using an Eddy SC 20 E-Beam Evaporator. PMMA coatings were placed in the evaporation chamber with Fe pellets, then sealed and held at high vacuum ($10^{-5}$ torr, 50 K). The cap thickness was monitored in-situ with a calibrated crystal balance. The resulting Fe capped Janus particle films had 100 nm coatings on the exposed hemispheres. Film substrates were cut into small sections and sonicated in ethanol for 10 s to remove the Janus particles. The ethanol was subsequently evaporated to leave dried magnetic Janus particles.

## Experimental set-up
See Supplementary Information (Supplementary Fig. 3) for full diagram of experimental set-up. 4 ×1000 Gauss neodymium magnets were attached to a motorized wheel at 90° intervals. The outward facing pole was alternated, 0° = N, 90° = S, 180° = N, 270° = S. The motorized magnet was supported on a height adjustable stand. A cuvette holder and a USB microscope were held in a fixed position. Rotation of the magnetic field is clockwise relative to the camera view during data acquisition. The field strength relative to orientation was measured experimentally using a gaussmeter.

## Sample preparation
Janus particles were suspended in a small amount of ethanol in a clean glass vial. A magnet was used to separate Janus particles from any uncoated PMMA particles as well as dust. The ethanol, uncoated particles and dust were removed using a glass pipette and replace with fresh ethanol. This process was repeated 4 times to ensure samples were clean. Janus particles were then dried in a weighing boat and subsequently 5 samples were prepared with increasing masses of Janus particles suspended in ethanol (2 mL).

## Responsive granular experiments
Before every sample loading, the magnet was centred at 0° (North). Samples were transferred to a glass cuvette and were subsequently loaded into the experimental apparatus and allowed to settle for 2 min at the maximum height of the set-up. Samples were then agitated with a vibrating motor to give a flattened granular bed. The distance measured was from the base of the cuvette (the thickness of the cuvette base is accounted for in all relevant calculations. Heights used in these experiments were $h$ = 25, 27.5, 30, 32.5, 35, 37.5, 40, 42.5, 45 and 60 mm from the magnet surface. $h_{max}$ = 60 mm. The rotation rate of the magnetic wheel was set to 1.2 rpm, equivalent to 2.4 Hz (cycles) (1 rotation = 4 N−S pole switches). When the magnetic rotation was activated, the system was allowed to reach a steadystate or until $3 \times 10^5$ ° of rotation was attained. Subsequently, a 10 srecording at steady-state was taken. All slope analysis and PIV was performed on the 10 s steady state recordings. The distance between the cuvette and the magnetic surface was the only parameter changed between measurements and samples, with cuvette position and orientation fixed for all experiments, and the motorized magnet moved further/closer with the use of an adjustable height stand.

## Analysis
Using ImageJ software, video recordings were converted to 8-bit and a 400 × 200 (w x h) pixel ROI was cropped parallel to the heap surface in the steady-state. These were saved as image sequences. Velocity flow fields were generated using PIVLab scripts in Matlab[2]. A region of width 10 vector units was averaged to give the velocity profiles for each sample at each magnetic field strength. $v_{max}$ was interpolated for each curve using a linear fit in the subsurface region of the flowing layer of each velocity profile.

# Data availability
The source data is presented in the supplementary datafile. Source data are provided with this paper.

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

## Acknowledgements

S.R.W.-W. and J.G. were supported by The Johns Hopkins University Applied Physics Laboratories and M.C.R. was supported by the National Science Foundation under Grant No. 1931681. The authors also acknowledge support from the McClurg Endowment Faculty Development Fund of the Department of Chemical and Biomolecular Engineering at Lehigh University and equipment usage in Lehigh's Institute for Functional Materials and Devices (I-FMD). We appreciate discussions with Richard Lueptow and Julio M. Ottino.

## Author contributions

S.R.W.-W. performed primary experimental work, analysis and co-composed the manuscript. J.F.G. co-composed the manuscript, supervised the research and performed analysis. M.C.R. contributed to analysis. W.E.B. contributed to preliminary experimental work. J.G. synthesised the Janus particles.

## Competing interests

The authors declare no competing interests.
