## [Peer Review File · Nature Communications]

Microrollers flow uphill as granular mediaREVIEWER COMMENTS

Reviewer #1 (Remarks to the Author):

Wilson-Whitford et al. describe an experimental study in which a rotating magnetic field applied to dry Janus particles in a packed bed causes them to roll uphill with a negative dynamic angle of repose. This method allows the authors to apply torques and friction to individual particles at field strengths above a threshold value. This is an interesting and unusual result and the velocity profiles are similar to the nonlocal properties of granular heaps. I think the results are exciting and have not yet been reported in the literature, with significant relevance to new technologies of particulate mixing which is of broad relevance to many fields. For these reasons of novelty and significance, I support publication in Nature Communications. The manuscript can be further improved if the authors address the following comments:

Granular bed depth and nonlocal rheology – Fig. 3 shows velocity profiles of the magnetic beads at different depths. The authors claim that they are similar to granular flows. Please provide a more quantitative comparison to granular flows. For example, how do these profiles compare to the nonlocal rheology models proposed by Kamrin and Bouzid? It would be important to at least examine the experimental data's match with existing models to support the claims of similarity to granular media.

Fig. 3 – There are negative velocities found at low bed depths. Do they represent backflow, and could these negative velocities correlated with confined flows? Please discuss.

Line 102 – Could the authors provide a more quantitative explanation for this statement? Or perhaps cite a study that looked at balance between driving forces for motion in a granular heap against opposing friction? Maybe they can also correlate this with their observation for the transition near $\phi = 1$.

Line 158 – The authors provide a qualitative explanation for a counterintuitive observation. Can the authors support their hypothesis by citing studies that might have looked at a similar observation?

Line 161 – Is there a way to replot Fig. 4 as a function of work done by beads instead of velocity, since there is an argument made in the manuscript about local dissipation through friction?

Line 175 – The authors shared an interesting observation in this paper which relates gravity driven flows with activated particle flow. Can they also elaborate on possible engineering applications of this phenomena?

There are multiple typos and grammatical errors in the manuscript that could use a more detailed lookthrough.

Reviewer #2 (Remarks to the Author):

The manuscript titled “Microrollers flow uphill as granular media” by Wilson-Whitford et al. demonstrates the experimental concept of uphill granular flow driven by an external force. The kinematics of the experiments presented here represent granular matter flow under gravity, but, in reverse. Authors accomplish the uphill flow using Janus microbeads that are responsive to an external magnetic field. The experiments and analysis performed provide new insights into bulk granular flows under external forcing. Authors observe that the (negative) angle of repose of the microroller system is characterized by two distinct regimes with respect to the applied magnetic field, demarcated by the system fluidization point. In addition, authors present a universal behavior for the system kinematics through collapse of the uphill velocity profiles, using a dimensionless shear rate.

The experimental results are robust. Nonetheless, the rest of the manuscript does not provide a good understanding of the underlying physical phenomena. I am troubled that the central theme of the manuscript remains unanswered – why does the uphill flow of microroller system (with its complex interactions) behave similar to the flow of granular matter under gravity? As such I have my reservations to publish the manuscript in its current form in Nature Communications. Please refer to my comments below for further details.

(1) The flow of granular matter under the influence of gravity and the associated flow regimes is well understood. My main question for the authors - how is the physics of the flow behavior obtained by uphill driven flow fundamentally different from the granular flow under gravity? In Fig 1a, authors showed the flow of microroller system under gravity. How do the three (or two) regimes: freely flowing, intermittent (may or may not be present), and the creep regimes (described in Komatsu et al. and other works), differ in both the systems (1a and 1b)? I do not want authors to perform more experiments but instead require a comparison of flow profiles of the existing data from 1a or others in the literature with the data presented here. I ask this since it is repeatedly mentioned in the text that the flow profiles are similar. Is it just qualitative?

(2) I feel that a more in-depth analysis will uncover the rich physics hidden in the existing velocity profiles in Fig 3. I am thinking the analysis on the lines of Houssais et al., where the authors mapped the flow profiles to respective flow regimes and proposed a phase diagram. Estimation of an inertial or viscous number for the microrollers in the suspension with external driving may give rise to a similar flow phase diagram for the system, which will depend on the driving force $(\beta/\beta_0)^2$ and the system dimensions $\Delta/2a$ reported in this manuscript. It would be curious to hear authors' thoughts about such an analysis.

(3) The abstract claims that the experimental system presented here has torque and friction (through Janus dipole interactions) independently applied. This is a bold claim. The dipole-based friction is an inherent property of the system due to the interactions with the solvent, ethanol. A simple system of microrollers climbing uphill, by itself, is not a well understood phenomena. An additional interaction parameter (in this case, from dipoles) further complicates the analysis of data (especially in Fig 2), as it becomes difficult to decouple the individual effects and the associated uphill flow behavior. I am trying to understand the two regimes in Fig 2, where the transition happens once the system is fluidized, especially with respect to the origins of the complex collective behavior presented in the graph and the interactions present in the system. Furthermore, simulations in dense suspensions (Singh et al.) discuss two types of friction that constraints the motion of particles in a shear flow: (a) the sliding friction and (b) rolling friction. The former comes from the interparticle friction restricting the sliding motion (here the origin of sliding-type of friction can come from the Janus particle surface roughness), while the latter arises due to constraint from interparticle interactions (which, in this case, comes from the dipole-dipole interaction which has attraction-like forces). Can the transition in Fig 2 be compared to change from one form of "frictional" behavior to another? The value of ϕ in the Fig 2 only refers to the particles in motion. Can any comments be made on whether they are rolling or sliding? Can authors expand further on the change in the angle of repose and the transition point of fluidization to the interactions involved in the system?

(4) I was wondering why a dimensionless shear rate was used to scale the vertical axis of the Fig 4, which represents the depth of the system from its free surface. One would assume an inherent length scale, as observed in earlier work with suspension flow along inclined plane (Bonnoit et al.), to be a more viable parameter to scale the y-axis. Why do authors think this is not the case? Moreover, the dimensionless shear rate presented here plateau off at $(\beta/\beta_0)^2 \approx 4$ (Fig 4b), which is around the minima of Fig 2. Is there any physical connection between this $(\beta/\beta_0)^2$ value in both the figures? What do authors think about this?

Minor Comments:

(1) It would be useful to include a schematic representation, a condensed version of the SI Fig 3 (preferably S3a), as a part of the main Fig 1, with the cuvette setup, position of the magnet, and the respective flow directions (uphill and recirculation) to guide the eyes of the reader.

(2) The supplementary figures (1, 2, 4, and 6) are not referred in the main manuscript. The authors should explain their relevance in the main manuscript (main text or the Methods section) and refer accordingly.

(3) The supplementary figure S1 (and in the Materials subsection in the Methods section in the main manuscripts) have notations denoted as $d[X,Y]$ (e.g. $d[3,2]$, $d[2,0]$ etc.). Authors should explain these notations in the text of the main manuscript.

(4) I did not understand the relevance of left y-axis in the supplementary figure S5 (h_{max2}/h_2 vs. h)? Please be careful about the wordings in the description for the same figure.

(5) I have the same question with Fig. S6, regarding its relevance (they are not explaining anything in the manuscript). Moreover, what is the significance of the computed Re number towards the end of SI document, since it is not mentioned anywhere the manuscript?

References:

Komatsu et al. "Creep Motion in a Granular Pile Exhibiting Steady Surface Flow", *Physical Review Letters* (2001).

Houssais et al. "Onset of sediment transport is a continuous transition driven by fluid shear and granular creep", *Nature Communications* (2015).

Singh et al. "Shear Thickening and Jamming of Dense Suspensions: The Roll of Friction", *Physical Review Letters* (2020).

Bonnoit et al. "Mesoscopic Length Scale Controls the Rheology of Dense Suspensions", *Physical Review Letters* (2010).

Reviewer #3 (Remarks to the Author):

This manuscript explores the behavior of a magnetic granular system, enabled by the ability to produce extremely high volumes of Janus magnetic particles. The chief result is that this system has a negative angle of repose, though surprisingly, velocimetry shows that bulk flow behavior is similar to a frictional granular system. This work is interesting and timely, and the experiments are thorough; the results are likely to resonate with several communities. I recommend publication in Nature Communications, provided the revisions outlined below are completed. In its current form, the manuscript is lacking some discussion points, and some of the figures could be made easier to interpret. Moreover, I believe the comparison with to granular systems could be strengthened (with existing data) by making some of the statements in the manuscript more quantitative - this would greatly enhance the impact of the manuscript. Below I outline specific points that I recommend are addressed before publication.

1. Friction is discussed throughout the manuscript, can more information be provided as to the source of this: fluid drainage vs. Columbic friction vs. magnetic interactions, for example? Is it possible to use a dimensionless number to tackle the estimate? If there is no appropriate dimensionless number, at the very least the strength of the magnetic interactions can be estimated as the iron thickness and the field strength have been measured. How does the magnitude of this interaction compare with the other interactions, and can this be used to perhaps strengthen the conclusion that the system behaves 'like' a granular flow? The manuscript could benefit from a more quantitative approach to characterizing friction, which should be feasible as I believe the authors indicate that the main source of friction is magnetic. I would push the authors to try and make more quantitative the intriguing observation that the flow strongly resembles granular flows.

2. The role of the fluid is unclear, and not covered even at a discussion level; it is not clear if a fluid is strictly necessary. Could the experiments be done completely in the absence of fluid? If fluid is needed, why is ethanol chosen, rather than say water or oil? The manuscript would benefit from some discussion on this topic, as it is could be the case that the fluid plays a role in determining particle friction in addition to the particle-particle magnetic interactions.

3. What is the individual particle rotation rate? Does it vary throughout the packing? It is stated that "Particles are not rotating in perfect synchronicity with the magnetic field due to the stress imposed by the weight of the granular bed and friction resulting from multiparticle interactions." Has this been measured? I could imagine this could be done with existing data either directly due to the Janus nature of the particles, or by looking at the FFT of the movies. Are the particles slipping and rotating erratically? Is there a variation in rotation frequency throughout the pack? The scaling collapse with dimensionless shear suggest that the rate cannot be too different from the field frequency, but some measurement of individual particle dynamics would strengthen the manuscript.

4. The terms force and torque appear to be used interchangeably, which adds a great deal of confusion for the reader. As far as I understand, the applied field is constant in magnitude and rotating in direction; there are no field gradients present across the sample. Thus, the particles are actuated via a magnetic torque, not a force, which is not how the manuscript describes it, for example "Below β_0 , the magnetic force on particles is insufficient to overcome static friction and the force network..." (line 62), "The driving force, a rotating set of magnets, both results in particle rotation.." (line 171), etc. I recommend that the language be changed.

5. I have some concerns about Figure 4. (a) Why is the strongest field strength excluded from the collapse? This is not clear from the Figure, as the square data points do not look markedly different than the other points (though there is no legend so I am unsure if the symbols are the same as Figure 3). Could the authors speculate about why this may be a failure point for the collapse - is this a different regime, are magnetic interactions too strong, etc.? (b) What are the lines on the figure - the slope one labeling is confusing, as this does not appear to be a log-log plot. Is the slope of this (presumably fitted) line meaningful to compare with known results from granular flows? It would be interesting to probe how far this analogy can be pushed (as discussed in point 2 above)

6. I have confusion regarding Figure 3. The manuscript states "The flow recirculation, described above, is represented by $v_x < 0$ and stronger for shallow bed depths where the entire bed is fluidised.", but if I understand the plots correctly this data is not shown. In all panels, $v_x \rightarrow 0$ at large depth. I would recommend additionally showing data for the fraction of the bed which is fluidized. I know that this information is indicated with a colormap in Figure 2, but this a very information-dense figure. I would recommend splitting into multiple panels to separately show the angle of repose and the fraction of the bed which is fluidized.

7. The statement "that behaves as the inverse bulk rheology and kinematics of a purely dissipative granular system" in the abstract does not really seem to be addressed in the manuscript - what is meant by 'inverse bulk rheology'? Is this related to the negative angle of repose?^{SEP}

7. Figure S6 needs more exposition. What exactly is the velocity be measured - is it a translational velocity? Does this velocity = ωR , e.g. are these particles 'wheels'? How does the single particle velocity compare to the velocities measured in the full packing? It is quite interesting to me that this is an overdamped system that seems to obey (presumably) inertial flow behavior that is seen in granular systems, and I think this point should be discussed in the main text (I would expect the Re to be comparable in the full system unless the velocities are markedly different). Additionally, I think the Re estimate is not quite correct, as the density of the particle should be used (but this will not significantly change things, it seems to make $Re \sim 3 \times 10^{-3}$).

8. The monochromatic figures (3,4) are very difficult to interpret as the symbols all appear nearly the same at such small sizes. If a colormap is not used, it would be very helpful to have more distinguishable symbols, for example making some hollow, using X's, +'s, etc. This is especially important for evaluating the strength (or lack thereof) of the collapse in Figure 4.

Our responses are in red.

REVIEWER COMMENTS

Overall description of friction in this manuscript:

To different degrees, all of the reviewers have included insightful questions about the role of friction in this system. First, the manuscript has been edited to clarify what we know and what we don't know about the role of friction. This is complicated because a traditional view of friction where the shear strength is related to the normal stress and cohesion through the angle of repose may or may not be appropriately quantified in this study. Nonetheless, the traditional manner of quantifying the angle of repose is included and more description of how the kinematics of the system lead to two maximum slopes in the opposite limits of magnetic strength as imposed by the distance of the magnets from the granular bed. In the limit of small magnetic field strength, this approaches the limit of the static angle of repose of the material. In the limit of the strong magnetic field strength, the angle of repose of the fully fluidized system also represents the pressure gradients within the recirculating bed. This is indicated in the manuscript. We hope future studies that include simulations will help understand this at the microscopic level, inspired by this manuscript that first demonstrates experimental evidence of uphill flow and heaping.

Response to Reviewer #1 (Remarks to the Author):

Wilson-Whitford et al. describe an experimental study in which a rotating magnetic field applied to dry Janus particles in a packed bed causes them to roll uphill with a negative dynamic angle of repose. This method allows the authors to apply torques and friction to individual particles at field strengths above a threshold value. This is an interesting and unusual result and the velocity profiles are similar to the nonlocal properties of granular heaps. I think the results are exciting and have not yet been reported in the literature, with significant relevance to new technologies of particulate mixing which is of broad relevance to many fields. For these reasons of novelty and significance, I support publication in Nature Communications.

We appreciate this thorough review and are encouraged by your feedback. Our comments are in red below:

The manuscript can be further improved if the authors address the following comments:

Granular bed depth and nonlocal rheology – Fig. 3 shows velocity profiles of the magnetic beads at different depths. The authors claim that they are similar to granular flows. Please provide a more quantitative comparison to granular flows. For example, how do these profiles compare to the nonlocal rheology models proposed by Kamrin and Bouzid? It would be important to at least examine the experimental data's match with existing models to support the claims of similarity to granular media.

We agree that a more quantitative agreement between our measured data and existing models would be advantageous. Unfortunately this is complicated by the fact that these models need both a measurement of local density and/or a measurement of the particle fluctuation. We did consider going deeper into our scaling analysis to understand how particle rotation relates to apparent fluctuations at the microscale, but this is difficult to confirm from our experimental results. In this manuscript we are only measuring the kinematics of the flow using PIV. We are planning to model these systems to get this level of information to compare with both theory and experiments. We feel the similarities in measured kinematics and the high-level scaling arguments bring us to a conclusion that these materials should, at the least, be analyzed as granular flows. We hope that other groups will follow our work and provide additional insight into the micromechanical properties of this system.

Fig. 3 – There are negative velocities found at low bed depths. Do they represent backflow, and could these negative velocities correlated with confined flows? Please discuss.

The negative velocities measured do represent backflow/recirculation and the description of this is made clearer in the manuscript. In those samples where the magnetic field is relatively weak, the heaping occurs in a limited region near the surface. There is almost no recirculation in these systems. Although we cannot do infinitely long experiments, the trend in the limit of a weak magnetic field is to have a static heap where the torque can no longer overcome the friction and potential energy to move upward. Alternatively, in the limit of strong magnetic fields, the system becomes fluidized and the stronger interparticle interactions drive a stronger uphill response that is truly dynamic and causes recirculation.

Line 102 – Could the authors provide a more quantitative explanation for this statement? Or perhaps cite a study that looked at balance between driving forces for motion in a granular heap against opposing friction? Maybe they can also correlate this with their observation for the transition near $\phi = 1$.

We have not provided a more quantitative explanation for this statement for the reason that this point in the graph of angle of repose indicates the transition between two regimes. On the left,

force chains supporting the granular bed may still exist and on the right of the point where $\phi = 1$ force chains no longer exist. It is complicated and essentially two completely different flow regimes. We hope to understand this further with paired simulations in future studies.

Line 158 – The authors provide a qualitative explanation for a counterintuitive observation. Can the authors support their hypothesis by citing studies that might have looked at a similar observation?

We have added citations to studies that studied the kinematics of flowing granular materials as a function of the cohesive properties of those materials. We stated that our result, in light of these studies, is counterintuitive because the opposite trend is found. It is not counterintuitive when considering that the stress driving the flow is imparted locally and cohesion is necessary to gain traction to have local deformation.

Line 161 – Is there a way to replot Fig. 4 as a function of work done by beads instead of velocity, since there is an argument made in the manuscript about local dissipation through friction?

We plotted the data in this way because it mirrored the granular dynamics literature. Friction/cohesion in the system is a necessary condition for driving granular flow. Because of this, we feel that plotting the shear rate as a function of the magnetic field strength is the most transparent and direct representation of this relationship. The work done by the beads could be interpreted multiple ways, either related to the torque imposed (which is not exactly known), by the difference between free flowing particles and those that are rolling uphill (which does not single out the role of static vs. cohesive interactions influencing friction), or solely related to the potential energy (angle of repose). All of these measurements would not be complete because of the changes in the local density that are not measured in these experiments. We hope to resolve these questions in future work where these experiments are paired with computational studies.

Line 175 – The authors shared an interesting observation in this paper which relates gravity driven flows with activated particle flow. Can they also elaborate on possible engineering applications of this phenomena?

We have several applications planned for this work (studies that include mixing, segregation, transport/conveying of material, and using particle beds for separations). We would rather have this first study identify the underlying physics and from there deviate toward application. Further fundamental and applied studies are in the works.

There are multiple typos and grammatical errors in the manuscript that could use a more detailed lookthrough.

We have reviewed the manuscript to remove additional typos and grammatical errors.

Response to Reviewer #2 (Remarks to the Author):

The manuscript titled “Microrollers flow uphill as granular media” by Wilson-Whitford et al. demonstrates the experimental concept of uphill granular flow driven by an external force. The kinematics of the experiments presented here represent granular matter flow under gravity, but, in reverse. Authors accomplish the uphill flow using Janus microbeads that are responsive to an external magnetic field. The experiments and analysis performed provide new insights into bulk granular flows under external forcing. Authors observe that the (negative) angle of repose of the microroller system is characterized by two distinct regimes with respect to the applied magnetic field, demarcated by the system fluidization point. In addition, authors present a universal behavior for the system kinematics through collapse of the uphill velocity profiles, using a dimensionless shear rate.

The experimental results are robust. Nonetheless, the rest of the manuscript does not provide a good understanding of the underlying physical phenomena. I am troubled that the central theme of the manuscript remains unanswered – why does the uphill flow of microroller system (with its complex interactions) behave similar to the flow of granular matter under gravity? As such I have my reservations to publish the manuscript in its current form in Nature Communications. Please refer to my comments below for further details.

Thank you for your thoughtful and thorough review of our manuscript. Granular dynamics, in themselves, are an active area of research. New models are proposed with regular frequency to understand their underlying physical phenomena. Likewise, research in active/responsive systems is progressing rapidly and reporting similarities to classic systems where the underlying physics is similar, yet unexplained (like turbulence). Thus, we are not answering the fundamental underpinnings of either granular dynamics nor those of this novel system. We are comparing the apparent dynamics of our system to granular systems and the similar complexities that arise, such as a defined angle of repose and a near surface flowing layer that scales by the imparted stress and demonstrating they scale in similar ways. We hope this work will stimulate future work in both the areas of granular systems and active/responsive systems and enable new insights and new approaches to quantifying their behavior.

(1) The flow of granular matter under the influence of gravity and the associated flow regimes is well understood.

We respectfully disagree with this assessment that the flow of granular media is well defined from a fundamental understanding. Only recently have scaling laws enabled engineering of granular flows across various flow regimes, and these scaling laws require further fundamental description of the micromechanical interactions that happen at the particle scale and localized force distributions. This is a very active area of research with new developments occurring on a regular basis. Quantitative insights are often made in simulations and experiments that enable validation of these insights are difficult to perform. These types of studies are regularly published in Nature Publishing Group. Our system defines a novel testbed for progress in this area.

My main question for the authors - how is the physics of the flow behavior obtained by uphill driven flow fundamentally different from the granular flow under gravity? In Fig 1a, authors showed the flow of microroller system under gravity. How do the three (or two) regimes: freely flowing, intermittent (may or may not be present), and the creep regimes (described in Komatsu et al. and other works), differ in both the systems (1a and 1b)? I do not want authors to perform more experiments but instead require a comparison of flow profiles of the existing data from 1a or others in the literature with the data presented here. I ask this since it is repeatedly mentioned in the text that the flow profiles are similar. Is it just qualitative?

The point of this manuscript is to state that the physics of the flow behavior of this uphill driven flow is similar to that of gravity driven flow. This is not obvious conjecture when considering a system where local torque is applied to each particle in the system. We have clarified that we are focused on a regime of granular flow most similar to the rolling/cascading regime found in tumblers where there is no flow intermittency. It would include the both "Regime II-Bed Load" and "Regime III-Creeping" as mentioned in the article by Houssais et al. (also mentioned in the revised manuscript) as is the case in many gravity-driven flows. We appreciate that fluid-sheared transport is also a system to which we should compare our results, perhaps in the lowest Shields' stress regime considered. It is difficult to make exact quantitative comparisons because the driving force for flow is drastically different. However, the ability to apply the same scaling that results in collapse of both this data and the velocity profiles of traditional granular flows is more than a qualitative approach.

(2) I feel that a more in-depth analysis will uncover the rich physics hidden in the existing velocity profiles in Fig 3. I am thinking the analysis on the lines of Houssais et al., where the authors mapped the flow profiles to respective flow regimes and proposed a phase diagram. Estimation of an inertial or viscous number for the microrollers in the suspension with external driving may give rise to a similar flow phase diagram for the system, which will depend on the driving force $(\beta/\beta_0)^2$ and the system dimensions $\Delta/2a$ reported in this manuscript. It would be curious to hear authors' thoughts about such an analysis.

We completely agree with this approach. The profiles below the linear regime are similar to that of Houssais et al. study and other papers where the profile decays exponentially. In this case, this analysis is complicated by the fact that there is recirculation and flow in the negative x-direction, though the stress may still decay in a similar manner. We feel that this analysis is appropriate for our in-depth follow up study.

(3) The abstract claims that the experimental system presented here has torque and friction (through Janus dipole interactions) independently applied. This is a bold claim. The dipole-based friction is an inherent property of the system due to the interactions with the solvent, ethanol. A simple system of microrollers climbing uphill, by itself, is not a well understood phenomena. An additional interaction parameter (in this case, from dipoles) further complicates the analysis of data (especially in Fig 2), as it becomes difficult to decouple the individual effects and the associated uphill flow behavior. I am trying to understand the two regimes in Fig 2,

where the transition happens once the system is fluidized, especially with respect to the origins of the complex collective behavior presented in the graph and the interactions present in the system. Furthermore, simulations in dense suspensions (Singh et al.) discuss two types of friction that constraints the motion of particles in a shear flow: (a) the sliding friction and (b) rolling friction. The former comes from the interparticle friction restricting the sliding motion (here the origin of sliding-type of friction can come from the Janus particle surface roughness), while the latter arises due to constraint from interparticle interactions (which, in this case, comes from the dipole-dipole interaction which has attraction-like forces). Can the transition in Fig 2 be compared to change from one form of “frictional” behavior to another? The value of ϕ in the Fig 2 only refers to the particles in motion. Can any comments be made on whether they are rolling or sliding? Can authors expand further on the change in the angle of repose and the transition point of fluidization to the interactions involved in the system?

We appreciate your interpretation of the dynamics in this system. It is complicated/complex. We aimed to quantify the kinematics of the flow in this study and we do not have data on individual particle rolling or sliding behaviour. I believe dissociating torque (or sum of the torques applied to each particle) and the applied torque that results from the tendency of a particle to align its dipole with the magnetic field is necessary in our description. This, in part, also alleviates your concerns about our description of friction. In our description of dipole-dipole interactions, please recognize this is not in the colloidal regime. Dipole-dipole interactions refer to the macroscopic dipole of the magnetic material on each particle that can be approximated by a single vector that is off center. We have added some more qualitative description about the transition point of fluidization. Overall, we have revised the manuscript to make these points clearer.

(4) I was wondering why a dimensionless shear rate was used to scale the vertical axis of the Fig 4, which represents the depth of the system from its free surface. One would assume an inherent length scale, as observed in earlier work with suspension flow along inclined plane (Bonnoit et al.), to be a more viable parameter to scale the y-axis. Why do authors think this is not the case? Moreover, the dimensionless shear rate presented here plateau off at $(\beta/\beta_0)^2 \approx 4$ (Fig 4b), which is around the minima of Fig 2. Is there any physical connection between this $(\beta/\beta_0)^2$ value in both the figures? What do authors think about this?

The dimensional shear rate in all cases is obtained by multiplying by a single scalar rotation rate since we did not report our explorations of different rotation rates (they didn't make any difference). By making variables dimensionless, readers have a broader understanding of how the scaling is applied in this system. Note that the slope of the velocity profile is 1 in Figure 4, suggesting that the scaling is appropriate. As for the correlation of the onset of the plateau and the point where the minimum slope and transition to a fully fluidized system occurs is one we considered, but we have not identified a clear physical connection between them. The work by Bonnoit et al. closely controls the volume fraction of particles in suspension, something that is not controlled nor measured in this system. The depth (thickness) of the flowing layer (the linear regime of the velocity near the surface) changes drastically, as is apparent in Fig. 2. This also could be a point of further analysis in a future study.

Minor Comments:

(1) It would be useful to include a schematic representation, a condensed version of the SI Fig 3 (preferably S3a), as a part of the main Fig 1, with the cuvette setup, position of the magnet, and the respective flow directions (uphill and recirculation) to guide the eyes of the reader.

We did consider including this diagram in the main text. In oral presentations, an animated version of this setup is very useful, however the static version of the image sometimes creates confusion of the resulting motion of the particles (which is opposite to the direction of the motorized cylinder). A colleague who gave feedback on the manuscript suggested that we remove it for clarity.

(2) The supplementary figures (1, 2, 4, and 6) are not referred in the main manuscript. The authors should explain their relevance in the main manuscript (main text or the Methods section) and refer accordingly.

We have addressed this concern in our revised manuscript.

(3) The supplementary figure S1 (and in the Materials subsection in the Methods section in the main manuscripts) have notations denoted as $d[X,Y]$ (e.g. $d[3,2]$, $d[2,0]$ etc.). Authors should explain these notations in the text of the main manuscript.

The description of this standardized notation for particle size distributions has been added to the figure caption.

(4) I did not understand the relevance of left y-axis in the supplementary figure S5 ($h_{max2}/h2$ vs. h)? Please be careful about the wordings in the description for the same figure.

This caption have been updated for clarity.

(5) I have the same question with Fig. S6, regarding its relevance (they are not explaining anything in the manuscript). Moreover, what is the significance of the computed Re number towards the end of SI document, since it is not mentioned anywhere the manuscript?

The point of this figure and calculation is to bring transparency to the motion of individual microrollers and the role of fluid inertia at the scale of the particle. Clearly individual particles move in phase with the magnet and the fluid motion is viscosity-dominated.

Response to Reviewer #3 (Remarks to the Author):

This manuscript explores the behavior of a magnetic granular system, enabled by the ability to produce extremely high volumes of Janus magnetic particles. The chief result is that this system has a negative angle of repose, though surprisingly, velocimetry shows that bulk flow behavior is similar to a frictional granular system. This work is interesting and timely, and the experiments are thorough; the results are likely to resonate with several communities. I recommend publication in Nature Communications, provided the revisions outlined below are completed. In its current form, the manuscript is lacking some discussion points, and some of the figures could be made easier to interpret. Moreover, I believe the comparison with granular systems could be strengthened (with existing data) by making some of the statements in the manuscript more quantitative - this would greatly enhance the impact of the manuscript. Below I outline specific points that I recommend are addressed before publication.

Thank you for your insightful comments. Our response and concurrent edits have strengthened the manuscript.

1. Friction is discussed throughout the manuscript, can more information be provided as to the source of this: fluid drainage vs. Columbic friction vs. magnetic interactions, for example? Is it possible to use a dimensionless number to tackle the estimate? If there is no appropriate dimensionless number, at the very least the strength of the magnetic interactions can be estimated as the iron thickness and the field strength have been measured. How does the magnitude of this interaction compare with the other interactions, and can this be used to perhaps strengthen the conclusion that the system behaves 'like' a granular flow? The manuscript could benefit from a more quantitative approach to characterizing friction, which should be feasible as I believe the authors indicate that the main source of friction is magnetic. I would push the authors to try and make more quantitative the intriguing observation that the flow strongly resembles granular flows.

As was requested by another reviewer, we have added additional analysis about the role of friction and how it relates to the angle of repose. It is clear that the permanent dipole of each cap is much smaller than the induced dipole-dipole interactions. While the dimensionless parameter we use, $(\beta/\beta_0)^2$, is more phenomenological in nature, it essentially scales the strength of the torque normal stress of the granular material. It is a convenient scaling for experimentalists and the shear rate collapses with this dimensionless quantity (inset of Fig.4). Future studies are planned to pair these findings with simulations and to consider the role of fluid drainage (or degree of fluidization) as a result of the responsive motion of these particles.

2. The role of the fluid is unclear, and not covered even at a discussion level; it is not clear if a fluid is strictly necessary. Could the experiments be done completely in the absence of fluid? If fluid is needed, why is ethanol chosen, rather than say water or oil? The manuscript would benefit from some discussion on this topic, as it could be the case that the fluid plays a role in determining particle friction in addition to the particle-particle magnetic interactions.

We have conducted some limited amounts of experiments in air. Generally, the results are similar. When we synthesize the particles, we disperse them in ethanol mainly due to its lower surface energy, which is irrelevant in this study. As the continuous phase, the viscosity will alter the lubrication between particles. Also, the buoyancy of the particles will be affected. Using a viscous fluid rather than air avoided potential issues of electrostatic buildup on the particles or container surfaces. In the flow regime studied (steady flow, fully submerged), we do not believe that will significantly change the results observed. These insights have been added into the manuscript. Further study is necessary to confirm this, and we plan to follow up this work with different viscosities and we have plans to conduct experiments in air for specific applications.

3. What is the individual particle rotation rate? Does it vary throughout the packing? It is stated that "Particles are not rotating in perfect synchronicity with the magnetic field due to the stress imposed by the weight of the granular bed and friction resulting from multiparticle interactions." Has this been measured? I could imagine this could be done with existing data either directly due to the Janus nature of the particles, or by looking at the FFT of the movies. Are the particles slipping and rotating erratically? Is there a variation in rotation frequency throughout the pack? The scaling collapse with dimensionless shear suggest that the rate cannot be too different from the field frequency, but some measurement of individual particle dynamics would strengthen the manuscript.

Absolutely, the individual particle rotation rate changes through the packing. The degree of fluidization, given in Fig. 2, indicates that particles are arrested due to the weight of particles above. We agree with your assessment that the scaling in the flowing layer works well because the particles are not influenced by the pressure of particles above. In the flowing layer, the particles will move closer to that of the magnetic field, though the collisional motion of the particles changes the motion of these particles. FFT analysis from the images is not possible to get information about the particle rotation (the caps scatter differently than the uncoated particle surface). We have slightly altered this statement for clarity.

4. The terms force and torque appear to be used interchangeably, which adds a great deal of confusion for the reader. As far as I understand, the applied field is constant in magnitude and rotating in direction; there are no field gradients present across the sample. Thus, the particles are actuated via a magnetic torque, not a force, which is not how the manuscript describes it, for example "Below β_0 , the magnetic force on particles is insufficient to overcome static friction and the force network..." (line 62), "The driving force, a rotating set of magnets, both results in particle rotation..." (line 171), etc. I recommend that the language be changed.

Changes have been made to be more consistent and clear.

5. I have some concerns about Figure 4. (a) Why is the strongest field strength excluded from the collapse? This is not clear from the Figure, as the square data points do not look markedly different than the other points (though there is no legend so I am unsure if the symbols are the same as Figure 3). Could the authors speculate about why this may be a failure point for the collapse - is this a different regime, are magnetic interactions too strong, etc.? (b) What are the

lines on the figure - the slope one labeling is confusing, as this does not appear to be a log-log plot. Is the slope of this (presumably fitted) line meaningful to compare with known results from granular flows? It would be interesting to probe how far this analogy can be pushed (as discussed in point 2 above)

The regime we have analyzed is one where there is a continuous flow with a relatively flat interface, similar to the rolling/cascading regime in a rotating tumbler. At the two limits of strong or weak magnetic field, the strong field produces stronger chaining and a non-flat interface of chains of particles marching uphill (to be studied more in the future), and in the weak field this scaling appears to work almost all of the way down to the point where the thickness of the flowing layer is just a couple of particles thick. You can see it in Fig. 3. We have added this information to the manuscript. The lines on the figure are now described in the caption.

6. I have confusion regarding Figure 3. The manuscript states "The flow recirculation, described above, is represented by $v_x < 0$ and stronger for shallow bed depths where the entire bed is fluidised.", but if I understand the plots correctly this data is not shown. In all panels, $v_x > 0$ at large depth. I would recommend additionally showing data for the fraction of the bed which is fluidized. I know that this information is indicated with a colormap in Figure 2, but this a very information-dense figure. I would recommend splitting into multiple panels to separately show the angle of repose and the fraction of the bed which is fluidized.

For some of the experiments, there is a significant flow reversal that is easily seen in the graph. Others have a very small degree of flow reversal, but there is always a small amount. With the scaled version of these graphs shown in Fig. 4, you can see the degree of flow reversal more clearly. As for the degree of fluidization in Fig. 2, we have added an additional graph in the SI (S7) for readers to see where this data is obtained.

7. The statement "that behaves as the inverse bulk rheology and kinematics of a purely dissipative granular system" in the abstract does not really seem to be addressed in the manuscript - what is meant by 'inverse bulk rheology'? Is this related to the negative angle of repose?

We agree this was confusing. We have revised the manuscript abstract for clarity.

7. Figure S6 needs more exposition. What exactly is the velocity be measured - is it a translational velocity? Does this velocity = ωR , e.g. are these particles 'wheels'? How does the single particle velocity compare to the velocities measured in the full packing? It is quite interesting to me that this is an overdamped system that seems to obey (presumably) inertial flow behavior that is seen in granular systems, and I think this point should be discussed in the main text (I would expect the Re to be comparable in the full system unless the velocities are markedly different). Additionally, I think the Re estimate is not quite correct, as the density of the particle should be used (but this will not significantly change things, it seems to make $Re \sim 3 \times 10^{-3}$).

We have added additional exposition of the measurements in S6. This is not the value we use in the scaling, and because the single particle rolling velocity is relatively constant, it would be a simple scalar substitution if it were used. This measurement is simply to show that the rolling velocity of a single Janus particle in dilute conditions on a flat substrate will be independent of the magnetic field strength and so will not affect heaping.

8. The monochromatic figures (3,4) are very difficult to interpret as the symbols all appear nearly the same at such small sizes. If a colormap is not used, it would be very helpful to have more distinguishable symbols, for example making some hollow, using X's, +'s, etc. This is especially important for evaluating the strength (or lack thereof) of the collapse in Figure 4.

There is a balance in symbol size where larger symbols overlap each other and smaller sizes are hard to see. We felt the data is monotonic and continuous in a way in each curve and the curve placement is monotonic such that it did not require additional demarcation. In Fig. 4, the collapse is so good that it would be almost impossible to use color or other symbols to see individual data sets (which is the aim of this graph).

REVIEWER COMMENTS

Reviewer #1 (Remarks to the Author):

The authors have addressed my earlier questions adequately. Although I would still have liked to see a more quantitative explanation of the uphill advection, understandably, simulations and theory are out of the scope of this paper which focuses on the novel uphill flows of the rotational Janus particles. I would be happy to recommend publication in Nature Communications.

Reviewer #2 (Remarks to the Author):

I believe that authors have answered my queries with respect to Fig 1, 2, and 3 to the best of their ability and the manuscript is improved considerably. Nonetheless, I believe that an in-depth physical interpretation of the complex system is needed for the work to be publishable in the Nature Communications. I am specifically referring to the data collapse in Fig 4, which is the most important result, where the behavior is analogous to the gravity driven flows. The system dynamics is novel and is relevant to the active granular matter community. The system chemistry is not a new finding and as such, a more detailed interpretation of the system dynamics associated with the collective motion (with respect to Fig 4 results) will strengthen the manuscript.

(1) I am not convinced with authors' interpretation of scaling parameters, especially the vertical axis. Yes, the scaling parameter is appropriate, since slope = 1 in the upward "flowing" regime. I believe there is a much more fundamental meaning to the parameter, γ . One can interpret the parameter as the ratio of average particle-level translational shear rate (v_x/y) and the particle-level rotational rate. The parameter determines the kinematics of the system, at microscopic scale. Do authors think this is right? If so, this is not clear from the manuscript text.

(2) As per authors, the result in Fig 4b is counterintuitive. It is not. If γ is interpreted as the ratio of particle-level translational to rotational shear rates, an increase in the magnetic field strength should decrease γ (Since $\gamma \sim 1/\Omega$ and Ω is directly proportional to the magnetic field strength). In addition, this gives a clearer answer to plateau-ing γ at higher magnetic field strengths. One can interpret this constant value as the minimum possible uphill collective motion in the system. This may be due to the increased chaining events or the finite surface area available for dipole-dipole interactions (system constraints), as authors explain in the paper.

(3) If authors agree with my interpretations above, then the y-axis in Fig 4 is a non-dimensional velocity in z-direction (non-dimensional particle-level shear rate x non-dimensional direction). I have two comments to improve the discussions in text in this context:

a. Why did the scaling change from $2a\Omega$ to v_{max} from Fig 3 to Fig 4? I can imagine the former not giving a good overall scaling in Fig 4. If that's the case why isn't Fig 3 scaled with v_{max} ?

b. This is the most important part. The above interpretations of γ and subsequently the vertical axis makes three regimes in Fig 4 clearer.

Regime 1: When $\gamma.v_x/y > 0$, slope is 1 and represents the flowing layer.

Regime 2: $0 > \gamma.v_x/y > -5$, slope small and resembles laminar flow in a pipe.

Regime 3: $\gamma.v_x/y < -5$, slope \sim infinity and there is no flow. Which demarcates a region below which there is no flow possible. (I cannot make out the symbols in the region to compare with field strengths from Fig 3; that is major problem throughout the manuscript, especially 3 and 4; color with varying levels of transparency is a good way to represent overlapping symbols).

(4) Even though each figure is discussed in detail, there is no overall discussion for the manuscript in its current form. If the focus is on showing the similarity of the system dynamics to the gravity-driven flows, the discussion should be focused on the possible confining forces and how it changes with

$(\beta/\beta_0)^2$? More importantly, how can you compare confining forces in your system to a regular granular heap flow in various regimes in 4, as you move from regime 1 to 3? These fundamental aspects, which can be answered with new interpretation of Fig 4 is currently missing.

Other comments:

(1) There are "flow" and "static" states of the system in the negative angle of repose in uphill leaping. Can these regimes be marked in Fig 2?

(2) The discussion of force chains appears when it is convenient and is not followed through the entire manuscript in a coherent manner across all the observed regimes. It only appears while discussing high magnetic field strength cases.

(3) Can a Re change be estimated in case of collective motion. Fig S6 estimated Re for a single roller under magnetic field. How much will the Re change in case of collective motion as a function of increase in magnetic field strength? Maybe estimating just for the free surface layer from the existing v_{sup} values would be interesting to the readers.

(4) Minor: $\{\dot{\gamma}\}$ appears inside Fig 4, close to y-axis of -2.

Reviewer #3 (Remarks to the Author):

I have carefully reviewed the outlined changes in the manuscript, and the revised manuscript addresses the chief concerns I had with the manuscript. The added discussion regarding the the role of the fluid and the quantification of 'magnetic friction', and as to why the strongest field was excluded from the collapse have enhanced the manuscript. I recommend it for publication in its current form.

I will restate my suggestion that was not acted upon to use a colormap in addition to symbol shape to illustrate the different data sets in Figures 3 and 4:

"The monochromatic figures (3,4) are very difficult to interpret as the symbols all appear nearly the same at such small sizes. "

My recommendation for publication is not contingent upon this change, but I strongly believe it would add clarity to these figures.

Dear Reviewers:

We appreciate the feedback received and truly feel our manuscript is significantly improved based on your inquiries. Related to the last round of reviews, our responses are in red text below.

Reviewer #1 (Remarks to the Author):

The authors have addressed my earlier questions adequately. Although I would still have liked to see a more quantitative explanation of the uphill advection, understandably, simulations and theory are out of the scope of this paper which focuses on the novel uphill flows of the rotational Janus particles. I would be happy to recommend publication in Nature Communications.

Thank you very much for your kind comments. We appreciate the affirmation for publication. We have added one more graph that does help connect the field strength with the observed phenomena.

Reviewer #2 (Remarks to the Author):

I believe that authors have answered my queries with respect to Fig 1, 2, and 3 to the best of their ability and the manuscript is improved considerably. Nonetheless, I believe that an in-depth physical interpretation of the complex system is needed for the work to be publishable in the Nature Communications. I am specifically referring to the data collapse in Fig 4, which is the most important result, where the behavior is analogous to the gravity driven flows. The system dynamics is novel and is relevant to the active granular matter community. The system chemistry is not a new finding and as such, a more detailed interpretation of the system dynamics associated with the collective motion (with respect to Fig 4 results) will strengthen the manuscript.

We are glad we were able to address your concerns with regard to Fig 1, 2, and 3 and improve the manuscript. Your thorough review has made the manuscript stronger and we appreciate your acknowledgement that our work is novel and relevant to the active matter and granular communities. We agree the system chemistry is somewhat irrelevant, almost by design to ensure this discovery will have robust applicability.

To your point of describing the dynamics of the system, we have taken a more “traditional” approach by first quantifying the kinematics of the system to connect with the granular literature. The description of dynamics in non-active granular flows remains an area of continued scientific exploration. In several discussions scanning many research papers, we have yet to strongly connect our system to existing literature that would shine light on the dynamics.

Friction is complicated and the additional orientation-dependent forces in this study resulting from the particle dipole give surprisingly similar results to those granular systems where friction is independent of orientation. Mohr-Coulomb Theory, $\tau = \sigma \tan(\theta) + c$, where shear stress is balanced by the normal stress multiplied by the coefficient of friction represented by the angle of the flowing layer and the cohesion in the system, adds a generalized continuum-level assessment of granular flows and has been connected in a newly added Fig. 5. Any further analysis of the specific details of how the magnetic field influences particle-particle dipole interactions would require particle-level detail from our experiments, something that remains experimentally elusive.

The data in Fig 4 is chosen intentionally to be pure and transparent in the sense that no additional scaling is added to generate the observed data collapse connecting between the dynamics of the system (magnetic field strength and rotation rate) and the resulting kinematics. Fig. 4b represents a result that naturally comes from this scaling.

We herein address the points below in parallel.

(1) I am not convinced with authors' interpretation of scaling parameters, especially the vertical axis. Yes, the scaling parameter is appropriate, since slope = 1 in the upward "flowing" regime. I believe there is a much more fundamental meaning to the parameter, γ . One can interpret the parameter as the ratio of average particle-level translational shear rate (v_x/y) and the particle-level rotational rate. The parameter determines the kinematics of the system, at microscopic scale. Do authors think this is right? If so, this is not clear from the manuscript text.

We agree on this connection between particle level translational shear rate and the rotation rate. We have altered the manuscript to reflect this more clearly.

(2) As per authors, the result in Fig 4b is counterintuitive. It is not. If γ is interpreted as the ratio of particle-level translational to rotational shear rates, an increase in the magnetic field strength should decrease γ (Since $\gamma \sim 1/\Omega$ and Ω is directly proportional to the magnetic field strength). In addition, this gives a clearer answer to plateau-ing γ at higher magnetic field strengths. One can interpret this constant value as the minimum possible uphill collective motion in the system. This may be due to the increased chaining events or the finite surface area available for dipole-dipole interactions (system constraints), as authors explain in the paper.

In all of these studies, the rotation rate of the magnetic field, Ω , is constant. Yes, as you state, collective motion of the particles results in rotation rates that are inversely proportional to the shear rate, however the driving mechanism for this is related to the magnetic field strength. We have made the manuscript more clear to highlight these finer aspects of this study.

As for chaining events, this is exactly what we mean by cohesion. Prompted by your interpretation and the other reviewers previous questions, we have added a Fig. 5 that demonstrates the rheological model (similar to that established in the literature) that fits our data.

(3) If authors agree with my interpretations above, then the y-axis in Fig 4 is a non-dimensional velocity in z-direction (non-dimensional particle-level shear rate x non-dimensional direction). I have two comments to improve the discussions in text in this context:

a. Why did the scaling change from $2a\Omega$ to v_{\max} from Fig 3 to Fig 4? I can imagine the former not giving a good overall scaling in Fig 4. If that's the case why isn't Fig 3 scaled with v_{\max} ?

Fig. 3 is raw data. $2a\Omega$ represents the dynamics input into the system whereas v_{\max} represents the resulting kinematics. This follows the approach used by prior studies (e.g. Taberlet et al. 2003) for transparency in scaling. We thought this scaling would be most transparent and align best with past analyses.

b. This is the most important part. The above interpretations of γ and subsequently the vertical axis makes three regimes in Fig 4 clearer.

Regime 1: When $\gamma \cdot v_x/y > 0$, slope is 1 and represents the flowing layer.

Regime 2: $0 > \gamma \cdot v_x/y > -5$, slope small and resembles laminar flow in a pipe.

Regime 3: $\gamma \cdot v_x/y < -5$, slope \sim infinity and there is no flow. Which demarcates a region below which there is no flow possible. (I cannot make out the symbols in the region to compare with field strengths from Fig 3; that is major problem throughout the manuscript, especially 3 and 4; color with varying levels of transparency is a good way to represent overlapping symbols).

We have modified the language of the manuscript to better identify these regimes. The most recent literature aims to avoid breaking the flow into "regimes" and consider a single constitutive equation that might describe all of the dynamics.

(4) Even though each figure is discussed in detail, there is no overall discussion for the manuscript in its current form. If the focus is on showing the similarity of the system dynamics to the gravity-driven flows, the discussion should be focused on the possible confining forces and how it changes with $(\beta/\beta_0)^2$? More importantly, how can you compare confining forces in your system to a regular granular heap flow in various regimes in 4, as you move from regime 1 to 3? These fundamental aspects, which can be answered with new interpretation of Fig 4 is currently missing.

We have added more discussion summarizing these findings, especially in light of added Fig. 5. This added analysis better answers the question of the role of the field strength.

Other comments:

(1) There are "flow" and "static" states of the system in the negative angle of repose in uphill leaping. Can these regimes be marked in Fig 2?

We have made the figure more clear about the role of fluidisation, however we feel it is disingenuous to draw a dotted line demarking two regimes of flow. This transition is most easily identified by the change in slope, which in this case mostly happens around the same point.

(2) The discussion of force chains appears when it is convenient and is not followed through the entire manuscript in a coherent manner across all the observed regimes. It only appears while discussing high magnetic field strength cases.

Fig. 5 has been added to better outline this point and the force chains are mentioned again for the fluidised regime.

(3) Can a Re change be estimated in case of collective motion. Fig S6 estimated Re for a single roller under magnetic field. How much will the Re change in case of collective motion as a function of increase in magnetic field strength? Maybe estimating just for the free surface layer from the existing v^x values would be interesting to the readers.

The value of v_{\max} is only a factor of about 3 and the point of the calculation is to demonstrate the order of magnitude and that the fluid inertia does not significantly contribute to the particle-scale dynamics observed in this study.

(4) Minor: $\{\dot{\gamma}\}$ appears inside Fig 4, close to y-axis of -2.

Corrected, thank you.

Reviewer #3 (Remarks to the Author):

I have carefully reviewed the outlined changes in the manuscript, and the revised manuscript addresses the chief concerns I had with the manuscript. The added discussion regarding the the role of the fluid and the quantification of 'magnetic friction', and as to why the strongest field was excluded from the collapse have enhanced the manuscript. I recommend it for publication in its current form.

I will restate my suggestion that was not acted upon to use a colormap in addition to symbol shape to illustrate the different data sets in Figures 3 and 4:

"The monochromatic figures (3,4) are very difficult to interpret as the symbols all appear nearly the same at such small sizes. "

My recommendation for publication is not contingent upon this change, but I strongly believe it would add clarity to these figures.

Thank you very much for your kind comments. We appreciate the affirmation for publication. We did graph these in different colors and because of the strong data collapse it does not add to the interpretation. The top-most symbols dominate the visual appearance of the graphs in the main region of interest. We also have solace in the fact that our raw data will accompany publication.

REVIEWERS' COMMENTS

Reviewer #2 (Remarks to the Author):

I believe that the authors have clarified my earlier concerns in the the recent version of the manuscript. I thoroughly enjoyed the link to $\mu(I)$ rheology and the analysis of confining pressure in the system. The experimental system is will be a game changer in the field of active granular matter. I am happy to recommend publication in Nature Communications.